# Immobilization of Lipase B from *Candida antarctica* on Magnetic Nanoparticles Enhances Its Selectivity in Kinetic Resolutions of Chiral Amines with Several Acylating Agents

**DOI:** 10.3390/life13071560

**Published:** 2023-07-14

**Authors:** Fausto M. W. G. Silva, József Szemes, Akan Mustashev, Orsolya Takács, Ali O. Imarah, László Poppe

**Affiliations:** 1Department of Organic Chemistry and Technology, Budapest University of Technology and Economics, Műegyetem rkp. 3, H-1111 Budapest, Hungary; macgyvergouveia@edu.bme.hu (F.M.W.G.S.); szemes.jozsef@edu.bme.hu (J.S.); mustashev.akan@edu.bme.hu (A.M.); takacs.orsolya@edu.bme.hu (O.T.); aal-alwani@edu.bme.hu (A.O.I.); 2Chemical Engineering Department, College of Engineering, University of Babylon, Hilla Babylon 5100, Iraq; 3Biocatalysis and Biotransformation Research Center, Faculty of Chemistry and Chemical Engineering, Babeş-Bolyai University of Cluj-Napoca, Arany János Str. 11, RO-400028 Cluj-Napoca, Romania; 4SynBiocat Ltd., Szilasliget u 3, H-1172 Budapest, Hungary

**Keywords:** biocatalysis, kinetic resolution, chiral amines, acylation, lipase B from *Candida antarctica*, magnetic nanoparticles, continuous-flow reactor

## Abstract

In lipase-catalyzed kinetic resolutions (KRs), the choice of immobilization support and acylating agents (AAs) is crucial. Lipase B from *Candida antarctica* immobilized onto magnetic nanoparticles (CaLB-MNPs) has been successfully used for diverse KRs of racemic compounds, but there is a lack of studies of the utilization of this potent biocatalyst in the KR of chiral amines, important pharmaceutical building blocks. Therefore, in this work, several racemic amines (heptane-2-amine, 1-methoxypropan-2-amine, 1-phenylethan-1-amine, and 4-phenylbutan-2-amine, (±)-**1a**–**d**, respectively) were studied in batch and continuous-flow mode utilizing different AAs, such as diisopropyl malonate **2A**, isopropyl 2-cyanoacetate **2B**, and isopropyl 2-ethoxyacetate **2C**. The reactions performed with CaLB-MNPs were compared with Novozym 435 (N435) and the results in the literature. CaLB-MNPs were less active than N435, leading to lower conversion, but demonstrated a higher enantiomer selectivity, proving to be a good alternative to the commercial form. Compound **2C** resulted in the best balance between conversion and enantiomer selectivity among the acylating agents. CaLB-MNPs proved to be efficient in the KR of chiral amines, having comparable or superior properties to other CaLB forms utilizing porous matrices for immobilization. An additional advantage of using CaLB-MNPs is that the purification and reuse processes are facilitated via magnetic retention/separation. In the continuous-flow mode, the usability and operational stability of CaLB-MNPs were reaffirmed, corroborating with previous studies, and the results overall improve our understanding of this potent biocatalyst and the convenient U-shape reactor used.

## 1. Introduction

Chiral primary, secondary, and tertiary amines are crucial building blocks for the pharmaceutical and agrochemical industries. The necessity for these constituents in their enantiomerically pure form in several drugs requires the development of cost-effective and long-lasting catalytic procedures for their asymmetric synthesis [1,2].

Biocatalytic processes, which use different forms of enzymes to perform efficient and selective synthetic transformations, are sustainable methods characterized by excellent activity and stereo-, regio-, and chemoselectivity [3]. Among the enzymes used as biocatalysts, lipases are highlighted due to a variety of desirable properties, such as easy availability, no need for cofactors, mild operating conditions, and high efficiency. Since many lipases exhibit excellent enantiomer selectivity that enables the generation of enantiopure amines through kinetic resolution (KR) and dynamic kinetic resolution (DKR) of racemates, lipases have generally been successfully used in asymmetric synthesis [4,5]. Lipase B from *Candida antarctica* (CaLB) is a highly effective and reliable biocatalyst of this class. However, like every other free enzyme, it can have several limitations, such as instability, nonreusability, and sensitivity to several reactants, which immobilization can circumvent [6]. The immobilized forms of this versatile biocatalyst have been used for the KR of chiral alcohols [7,8,9,10], azolides [11], and amines [12,13,14,15,16,17].

At an industrial scale, CaLB and other lipases have been extensively investigated and used since the beginning of the 1990s for the enantioselective hydrolysis/formation of racemic esters and amides. In the mid-1990s, Bayer established the KR of α-methylbenzylamine. Even though a significant enantiomeric excess was achieved, the technique was considered unusable for larger scales due to the requirement of high catalyst concentrations. By the end of the 1990s, methods for enzyme immobilization were more developed, which increased their stability and allowed them to be recycled, making the procedure more effective. After some advances, such as discovering a particular acyl donor, ethyl methoxyacetate, BASF began using lipase-catalyzed KRs to create tons of specific chiral amines [18,19]. The two crucial steps in this process were to select a suitable acylating agent and efficient support for enzyme immobilization.

In the lipase-catalyzed stereoselective *N*-acylations, a variety of acyl donors (or the so-called acylating agents) can be used. Nonactivated esters of acetic acid, such as ethyl acetate [20,21]; isopropyl acetate [22]; *n*-butyl acetate; or other fatty acids, like myristic acid [23]; as well as the so-called activated esters, such as alkyl 2-alkoxyacetates and alkyl 2-cyanoacetates, including isopropyl 2-methoxyacetate [24], diisopropyl malonate [14], isopropyl 2-cyanoacetate [15], or isopropyl 2-ethoxyacetate [16,17], have already been applied. In the activated esters, an electron-withdrawing group (such as an alkoxy, cyano, or halogen) at the β-position amplifying the partially positive nature of the ester carbonyl group enhances the reactivity of such esters as acylating agents. Applying isopropyl esters instead of ethyl esters increases the apparent selectivity of the enzymatic process due to diminishing the nonselective chemical acylation occurring besides the lipase-catalyzed process.

Another advantage of malonates or cyanoacetates as acylating agents is the possibility of further molecule modifications, which is important in the pharmaceutical industry [14,15,16,17]. For example, from the product of Knoevenagel condensation of the enantiopure amide of (*S*)-1-phenylethan-1-amine and cyanoacetic acid, tyrphostins, which are inhibitors of protein tyrosine kinases that can be used as chemotherapeutic agents, can be synthesized [15]. The amides of other amines, such as (*±*)-1-methoxy-2-propanamine, (*±*)-1-phenylethan-1-amine, and (*±*)-4-phenylbutan-2-amine with isopropyl 2-cyanoacetate, can be used to produce aminoisoxazole derivatives, and they are effective due to altering protein kinase activity in treating diseases like cancer, cell proliferative disorders, Alzheimer’s disease, auto-immune diseases, and neurological disorders [25].

The type of support and enzyme immobilization technique also influences the effectiveness of the biocatalytic process. In general, immobilization may improve enzyme activity; stability; selectivity and/or specificity; and resistance to inhibitors, solvents, and chemicals. Apart from the enhancement of catalytic properties, the reusability of the biocatalysts may also be improved [26]. Lipases immobilized in/on various carriers (or hybrid catalysts, involving metal and an enzyme) have been used in KRs or DKRs in order to obtain enantiopure amines such as mesoporous beads [15,17], phenyl-functionalized silica gel [15,16], macroporous acrylic resin [14,15], siliceous mesocellular foams (MCFs) [27,28], metal–organic frameworks (MOFs) [29], and hollow structure materials [30].

Most frequently, porous supports are applied, which are advantageously prepared and applied but can suffer from several problems. One of the most utilized biocatalysts is the commercial Novozym 435 (N435), made via the immobilization of CaLB on macroporous Lewatit VP OC 1600 resin via adsorption and interfacial activation [31]. Some of the intrinsic problems simply stem from the adsorptive immobilization, enabling enzyme desorption under aqueous conditions, and material transfer limitations within the pores. Additional challenges that are unique to the support include mechanical brittleness, a moderate hydrophilicity that allows hydrophilic molecules (such as water or glycerin) to accumulate and leads to the most serious issue: disintegration in various organic mediums [31].

A possible solution to tackle this problem is using nonporous materials such as magnetic nanoparticles (MNPs), providing a large surface area and high surface-to-volume ratio for enzymes on the supports, thereby reducing diffusion limitation issues, which makes them appealing for industrial use [32]. Their superparamagnetic properties allow magnetic decantation, so MNPs can be easily removed from the reaction mixture [32,33]. Apart from the utilization in the KR or DKR of enantiomer compounds, CaLB-MNPs are also used in biodiesel synthesis [34,35], transesterification reactions [36], or even utilized in a magnetic crosslinked inclusion body form [37]. But, to the best of our knowledge, there are no reports of the utilization of MNPs in the KR or DKR of amines, except in one study with a magnetic hybrid catalyst containing lipase and palladium applied in the DKR of (±)-1-phenylethan-1-amine [38]. Therefore, it is a topic yet to be explored.

Continuous-flow production is gaining popularity in diverse industries, including in the pharmaceutical sector, with the main benefits of this cutting-edge technology being the capacity to execute chemistry experiments that are challenging to conduct in batch mode, as well as faster and safer reactions that may be more ecologically friendly [39]. Numerous researchers combined the advantages of MNP-immobilized biocatalysts and microreactor technology. In recent studies, a simple but effective and convenient MNP-based PTFE tube microreactor with six adjusting permanent magnets underneath the tube was used for the enantiomer selective acylation of drug-like racemic alcohols such as 4-(morpholin-4-yl)butan-2-ol [8], 4-(3,4-dihydroisoquinolin-2(1*H*)-yl)butan-2-ol, and 4-(3,4-dihydroquinolin-1(2*H*)-yl)butan-2-ol [9], catalyzed using a biocatalyst obtained via the covalent immobilization of lipase B from *Candida antarctica* on magnetic nanoparticles (CaLB-MNPs) in a continuous-flow mode. To broaden the spectrum of substrates evaluated in this system, a U-shape reactor was used, and reactions with selected chiral amines were performed. To enhance the accuracy and controllability of the magnetic U-shape reactor, a heat-insulating box was applied to perform reactions with temperature control (Figure 1).

Although CaLB has been recently studied for immobilization onto nonporous MNPs, due to having advantages over porous supports, there is a lack of studies on their behavior in the KR of racemic amines. This work reports the kinetic resolution of several chiral amines catalyzed with CaLB-MNPs utilizing different isopropyl esters as acylating agents in batch and continuous-flow modes.

## 2. Materials and Methods

### 2.1. Enzyme and Materials

Recombinant lipase B from *Candida antarctica* (CaLB, produced by microbial fermentation in *Pichia pastoris*, exhibiting a single band around 33 kDa on SDS gel EF; provided as a lyophilized powder, Lot-NO: MA-b-0002, activity: 59,900 TBU/g) for immobilization experiments was obtained from c-LEcta (Leipzig, Germany). All other reagents and solvents were products of Sigma-Aldrich (Saint Louis, MO, USA), Merck (Darmstadt, Germany), or Alfa Aesar Europe (Karlsruhe, Germany) and are listed in Appendix A.

### 2.2. Methods

The analytical methods and calculations utilized in this study were based on previously published methods [14], with minor modifications, as detailed below.

#### 2.2.1. Analytical Methods and Calculations

TLC was carried out using Kieselgel 60 F254 (Merck) sheets. Spots were visualized under UV light (Vilber Lourmat VL-6.LC, 254 nm) or after treatment with a 5% ethanolic phosphomolybdic acid solution and heating the dried plates.

Gas chromatographic (GC) analyses were performed with an Agilent 4890 gas chromatograph equipped with a flame ionization detector (FID) using H_2_ as carrier gas (injector: 250 °C, detector: 250 °C, head pressure: 12 psi, split ratio: 50:1) and a Hydrodex β-6TBDM column (25 m × 0.25 mm × 0.25 μm film with heptakis-(2,3 di-*O*-methyl-6-*O*-*t*-butyldimethyl-silyl)-β-cyclodextrin; Macherey and Nagel (Düren, Germany)) using the temperature programs indicated in Appendix A). GC chromatograms are given in Appendix A.

Conversion (*c*) and enantiomeric excess (*ee*) values were determined via GC utilizing previously published equations, with molar response factors 1.22, 1.06, and 1.185 for amides with the different acylating agents **2A**–**C**, respectively. The enantiomeric ratio/selectivity (*E*) was calculated from the enantiomeric excess of the substrate (*ee*_s_) and product (*ee*_p_) fractions using a previously published equation for irreversible KRs. Due to sensitivity to experimental errors, *E* values calculated in the 100–200 range are given as >100, those in the 200–500 range are denoted as >200, and those above 500 are denoted as ≫200 [40,41].

#### 2.2.2. Preparation of Isopropyl 2-Ethoxyacetate **2C**

2-Ethoxyacetic acid and isopropyl 2-ethoxyacetate **2C** were prepared using previously published methods [16], with minor modifications, as detailed below.

##### 2-Ethoxyacetic Acid

Sodium (10 g, 0.434 mol) was dissolved in water-free ethanol (450 mL) at 60 °C, and then triethylbenzylammonium chloride (TEBA-Cl, 1 g, 0.036 mol) and sodium 2-chloroacetate (40 g, 0.344 mol) were added to the solution. After refluxing the resulting mixture for 24 h, ethanol was removed under vacuum. The residue was dissolved in distilled water (400 mL), and pH was set to 2 via the addition of 5N HCl. After extracting the solution with ethyl acetate (3 × 100 mL), the combined organic phases were washed with brine (100 mL) and dried over anhydrous Na_2_SO_4_. After the removal of the solvent via vacuum rotary evaporation, the residue was purified with vacuum distillation to yield 2-ethoxyacetic acid as a colorless oil.

Yield = 76%; Bp = 90 °C (34 Torr); TLC: Rf = 0.29 (CH_2_Cl_2_:MeOH = 10:1); ^1^H-NMR (500 MHz, chloroform-*d*_3_, δ ppm): 8.92 (s, 1H), 4.15 (s, 2H), 3.65 (q, *J* = 7.0 Hz, 1H), 1.29 (t, *J* = 7.0 Hz, 2H); ^13^C-NMR (126 MHz, chloroform-*d*_3_, δ ppm): δ 175.15, 67.44, 40.57, 14.91; IR (ν cm^−1^): 2980, 2900, 1724, 1430, 1350, 1205, 1110, 1031, 1008, 877, 846, 669.

##### Isopropyl 2-Ethoxyacetate **2C**

2-Ethoxyacetic acid (28 g, 0.272 mol, 25.4 mL) and *p*-toluenesulfonic acid (0.92 g, 5.4 mmol) were dissolved in 2-propanol (150 mL), and the resulting mixture was refluxed for 32 h. During distilling off the excess alcohol at atmospheric pressure, an additional 100 mL of 2-propanol was added dropwise to perfectly remove the formed water via azeotropic distillation. The residual crude ester was purified via vacuum distillation to give pure isopropyl 2-ethoxyacetate as a colorless oil.

Yield: 15.1 g (81%); bp = 70 °C (38 Torr); TLC: Rf = 0.78 (hexane:EtOAc = 7:3); ^1^H-NMR (500 MHz, chloroform-*d*_3_, δ ppm): δ 5.11 (m, 1H), 4.04 (s, 2H), 3.60 (q, *J* = 7.0 Hz, 2H), 1.30–1.23 (d+t, 9H); ^13^C-NMR (126 MHz, chloroform-*d_3_*, δ ppm): δ 170.11, 68.38, 68.27, 67.11, 21.80, 14.99; IR (ν cm^−1^): 2979, 2876, 1747, 1375, 1275, 1206, 1136, 1104, 1031, 938, 855, 819, 727, 584.

#### 2.2.3. Preparation of CaLB-MNP Biocatalysts

The preparation of the MNP carriers, including the preparation of the magnetite core and its surface modifications, and further immobilization of CaLB were performed using previously published methods [8].

The surface morphology of the samples was investigated with a JEOL JSM-5500LV scanning electron microscope (SEM). An electron beam energy of 15 kV was used for the investigations. The SEM images are given in Appendix A.

#### 2.2.4. Kinetic Resolution of the Racemic Amines (±)-**1a**–**d** in Batch Mode with Different Acylating Agents **2A**–**C** and CaLB on Different Supports

In a 4 mL screw-cap vial, immobilized CaLB enzyme (10.0 mg, CaLB-MNPs or CaLB N435) and methyl *tert*-butyl ether (MTBE, 200 μL) were added. After sonicating the mixture for 10 min, the corresponding amine ((±)-**1a**–**d**, 12.5–100 mM) and the acylating agent (**2A**–**C**, 12.5–100 mM, 1 equiv.) were added. The reaction mixture was shaken at the rate of 200 rpm for 6 h at 40 °C and monitored by taking samples after different reaction times (1, 2, 4, and 6 h).

The samples (20 μL) from KRs were diluted with ethyl acetate (480 μL), treated with acetic anhydride (10 μL) at 25 °C and 350 rpm for 30 min in a shaker (for the derivatization of unreacted amines **1a**–**d** into their *N*-acetamides **1*a**–**d**) and quenched with 20 μL of distilled water. After drying over anhydrous Na_2_SO_4_, the samples were analyzed using GC on a chiral column. A variance of less than 2% was found in the preliminary activity test in batch mode, performed in triplicate. Therefore, the optimization and time course experiments were carried out as single series.

#### 2.2.5. Kinetic Resolution of the Racemic Amines (±)-**1a,b,d** with Isopropyl 2-Cyanoacetate **2B** in Batch Mode Using CaLB-MNPs for Isolation of the New Amides (R)-**3(a,b,d)B**

In a 4 mL screw-cap vial, racemic amine (±)-**1a,b,d** (57.6 mg, 44.5 mg, or 74.6 mg, respectively; 0.5 mmol), CaLB-MNPs (20 mg), and isopropyl 2-cyanoacetate **2B** (0.5 mmol, 1 equiv.) were added to MTBE (200 μL), and the resulting suspension was shaken at the rate of 200 rpm for 4 h at 40 °C in a Vibramax 100 shaker (Heidolph, Schwabach, Germany). The supernatant was decanted by anchoring the CaLB-MNPs with a neodymium magnet, and the resuspended CaLB-MNPs were washed using a mixture of methyl *t*-butyl ether (MTBE)–hexane (1:2 ratio, 2 × 2.5 mL). The washed CaLB-MNPs were dried in a fume hood overnight at room temperature and stored in a refrigerator until further use. From the combined reaction mixture and washing solutions, the volatiles were removed via vacuum rotary evaporation, and the resulting products were separated using preparative thin layer chromatography (silica gel, eluted with CH_2_Cl_2_:MeOH = 20:1) to yield the (*R*)-amide (*R*)-**3(a,b,d)B** and the residual (*S*)-amine (*S*)-**1a,b,d**; respectively.

##### 2-Cyano-*N*-(2-heptanyl)acetamide (*R*)-**3aB**

Yield: 29 mg, 31 % (*ee* = 99.5%, by GC); ^1^H NMR (500 MHz, chloroform-*d*_3_, δ ppm): 8.62 (s, 1H), 4.00 (p, J = 7.0 Hz, 2H), 3.38 (s, 2H), 1.54–1.43 (m, 2H), 1.39–1.26 (m, 6H), 1.20 (d, J = 6.6 Hz, 2H), 0.91 (t, J = 6.5 Hz, 3H); ^13^C NMR (126 MHz, chloroform-*d*_3_, δ ppm): 159.93, 114.90, 46.66, 36.52, 31.55, 25.96, 25.60, 22.52, 20.62, 13.98; IR (ν cm^−1^): 3282, 2964, 2951, 2923, 2850, 1660, 1555, 1469, 1453, 1357, 1332, 1245, 1078, 930, 720, 582.

##### 2-Cyano-*N*-(1-methoxy-2-propanyl)acetamide (*R*)-**3bB**

Yield: 38 mg, 48% (*ee* = 97.9%, by GC); ^1^H NMR (500 MHz, chloroform-*d*_3_, δ ppm): 8.85 (s, 1H), 4.18 (m, 1H), 3.45 (dd, J = 9.6, 3.9 Hz, 2H), 3.39 (s, 5H), 1.27 (d, J = 3.9 Hz, 3H); ^13^C NMR (126 MHz, chloroform-*d*_3_, δ ppm): 160.43, 114.63, 74.82, 59.15, 46.05, 25.98, 17.34; IR (ν cm^−1^): 3308, 2976, 2931, 1668, 1455, 1393, 1367, 1136, 1022, 845, 816.

##### 2-Cyano-*N*-(4-phenyl-2-butanyl)acetamide (*R*)-**3dB**

Yield: 28 mg, 26% (*ee* = 97.9%, by GC); ^1^H NMR (500 MHz, chloroform-*d*_3_, δ ppm): 8.85 (s, 1H), 7.31 (t, J = 7.4 Hz, 2H), 7.21 (m, J = 6.7 Hz, 3H), 4.07 (m, 1H), 3.32 (s, 2H), 2.69 (td, J = 7.6, 4.0 Hz, 2H), 1.85 (q, J = 7.5 Hz, 2H), 1.24 (d, 3H); ^13^C NMR (126 MHz, chloroform-*d_3_*, δ ppm): δ 160.18, 141.20, 128.56, 128.27, 126.12, 114.79, 46.57, 37.96, 32.43, 25.90, 20.72; IR (ν cm^−1^): 3306, 2974, 2929, 1665, 1545, 1454, 1393, 1367, 1136, 1022, 846, 700.

#### 2.2.6. Design and Assembly of the Thermostatted U-Shape MNP Reactor

The U-shape MNP reactor system (Figure 1), similar to its nonthermostatted predecessor [9], was designed using AutoCAD (2020 student version) and printed with a Rankfor100 3D printer (CEI Conrad Electronic International, Ltd., New Territories, Hong Kong). In this study, we used neodymium disc magnets, 4 mm × 2 mm, N35 (Euromagnet Ltd., Budapest, Hungary) as permanent magnets and a polytetrafluoroethylene (PTFE) tube with ID 1.50 mm as the reactor body and connection parts. The U-shape reactor was a holder allowing adjustable positioning of six permanent magnets under the PTFE tube in the closest possible vicinity to the PTFE tube reactor, thereby creating six sites where the magnets could anchor the CaLB-MNP biocatalysts inside the tube (Figure 1). A SpinSplit continuous-flow syringe pump (SpinSplit Technical Research and Development LLC, Budapest, Hungary) equipped with two glass syringes of 0–5 mL volume was coupled to the tubular reactor part of the reactor module. For heat control, a heat exchanger (water heat exchanger with aluminum heat radiator/heat sink for computer, size: 60 × 60 × 12 mm, 25 mL capacity) with a fan (size: 120 × 120 × 25 mm, 12 V DC) was connected to a Lauda Ecoline RE104 Recirculating Chiller as the heat source. The device was placed into a Styrofoam heat insulating box (270 × 235 × 145 mm internal dimension; 25 mm wall thickness), which was properly punctured for providing the inlet and outlet of the PTFE tube reactor and to allow for the installation of the tubes of the external chiller and electric wires for fan, with a Plexiglas cover (250 × 215 mm, 4 mm thickness). Pictures showing the entire reactor setup are presented in Appendix A.

#### 2.2.7. Kinetic Resolution of the Racemic Amines (±)-**1b** and (±)-**1c** with Isopropyl 2-Ethoxyacetate **2C** Using CaLB-MNPs in Continuous-Flow U-Shape Reactor

The U-shape reactor was used for the continuous-flow kinetic resolution of the designed substrate amines (±)-**1b** and (±)-**1c**. For filling the PTFE reactor tube of 1.50 mm ID as the reactor body, 6 × 1 mg of CaLB-MNPs were suspended in 6 × 1 mL of a mixture of methyl *t*-butyl ether (MTBE)-hexane (1:2 ratio), and the suspended CaLB-MNPs were supplied at a 50 µL min^−1^ flow rate into the reactor’s six chambers in a counter-current direction of the later fluid flow, filling the last chamber first and the first chamber last (by placing the permanent magnet under the tube in the actual position, from the last-to-first position).

Experiments were performed by pumping solutions of the racemic 1-methoxy-2-propanamine (±)-**1b** (100 mM) or 1-phenylethan-1-amine (±)-**1c** (100 mM), and isopropyl 2-ethoxyacetate (1 equiv.) in MTBE using the U-shape reactor filled with CaLB-MNPs at a 1 µL min^−1^ flow rate at 40 °C.

## 3. Results and Discussion

The first goal of this work was to carry out the CaLB-MNP-catalyzed kinetic resolution (KR) of four distinct racemic aliphatic and aromatic primary amines (heptane-2-amine, 1-methoxypropan-2-amine, 1-phenylethan-1-amine, and 4-phenylbutan-2-amine, (±)-**1a**–**d**, respectively) utilizing three different acylating agents in the batch mode: diisopropyl malonate **2A**, isopropyl 2-cyanoacetate **2B**, and isopropyl 2-ethoxyacetate **2C** (Figure 1). The reactions were also performed with the Novozym 435 (N435) form of the CaLB biocatalyst under the same conditions to compare the performance of the two forms of CaLB in terms of productivity and enantiomer selectivity.

### 3.1. Biocatalyst Characterization

CaLB-MNPs produced with CaLB immobilized covalently onto the MNPs were proposed as good biocatalysts for KRs of racemic compounds. To prove that the immobilization was successful, the protein concentration of the supernatant before and after immobilization was determined using the Bradford assay. The experiments indicated almost full immobilization (*Y* = 98%) after 120 min at a CaLB:MNP ratio of 1:15 under the conditions utilized in our previous work [8].

SEM images could not reveal the real surface morphology of the CaLB-MNPs due to the aggregation of the distinct nanoparticles (Appendix A). Consequently, the difference between the MNP carrier and CaLB-MNPs could be only revealed using indirect methods based on the activity measurement of the CaLB-MNPs and immobilization yield. Furthermore, the successful immobilization of CaLB on bisepoxide-activated macroporous resins [42] and aspartase on bisepoxide-activated MNPs [43] using the same chemistry support the conclusions drawn from indirect methods.

### 3.2. Kinetic Resolution of Chiral Amines (±)-**1a**–**d** with Different Acylating Agents (**2A**–**C**) in Batch Mode with CaLB-MNPs and N435

Similar reaction conditions to those previously described for the KR of all racemic amines (±)-**1a**–**d** with all acylating agents (**2A**–**C**) were explored to optimize the KR processes in this work [14,15,16,17]. The first screening reactions took place at 40 °C for 6 h in MTBE, toluene, and a mixture of THF:toluene 1:1 as solvents at 12.5 mM substrate concentration. This solvent screen revealed MTBE as the most efficient solvent due to the significantly lower conversions in the other two solvents tested, and it is also the greenest alternative. Therefore, MTBE was used as a solvent in all subsequent tests (Figure 1).

The results of the KRs in the batch mode at 45 mM substrate concentration are presented in Figure 2. This experiment was performed as detailed in Section 2.2.4, and the results were analyzed with GC on a chiral column by taking samples from the reaction at different intervals. The residual amines (*S*)-**1a**–**d** in KR mixtures were derivatized to their acetamides via Ac_2_O treatment to determine the conversion and enantiomeric excess of the components.

It is clear from Figure 2 that the various starting amines (±)-**1a**–**d** were converted in KRs with quite different rates and levels of selectivity. In the CaLB-MNP-catalyzed KRs, the fastest reaction and highest conversion were attained during the *N*-acylation of the racemic 1-methoxy-2-propanamine (±)-**1b**, with all acylating agents **2A**–**C** (Figure 2a,c,e). In the CaLB N435-catalyzed KRs, the acylation of racemic 1-methoxy-2-propanamine (±)-**1b** utilizing diisopropyl malonate **2A** (Figure 2b) and isopropyl 2-ethoxyacetate **2C** (Figure 2f) were faster. The CaLB-catalyzed acylation of the larger amines (±)-**1a,c,d** was slower usually but provided proper conversions (≥40%) after 6 h for all cases except when utilizing CaLB-MNPs and **2A** as the acylating agent (Figure 2a). The conversions were usually higher in CaLB N435 experiments, often surpassing the 50% theoretical conversion limit for KRs of exclusive enantiomer selectivity.

Regarding the enantiomeric excess of the products (*R*)-**3(a**–**d)(A**–**C)**—an indicator of enantiomer selectivity—the values for the CaLB-MNP-catalyzed KRs fluctuated above 80% during the experiments, usually reaching values of ≥99%. On the other hand, in several CaLB N435-catalyzed KRs, somewhat lower enantiomeric excess values could be achieved, e.g., 40% for (*R*)-**3bC** (Figure 2f). In summary, with CaLB N435, KRs were faster and reached higher conversion rates, but the enantiomer selectivity was lower than with CaLB-MNPs exhibiting higher selectivity with sufficiently good-to-excellent conversions.

Considering all the experiments shown in Figure 2, it can be concluded that utilizing CaLB-MNPs as biocatalysts and isopropyl 2-ethoxyacetate **2C** as the acylating agent (Figure 2e) provides the best balance between conversion and enantiomer selectivity. In this case, *N*-acylation was accomplished with a good conversion value (50%) for amines (±)-**1a,b,d** even after 1 h, while the KR of 1-phenylethan-1-amine (±)-**1c** reached this conversion after 6 h. The enantiomeric excess for amides (*R*)-**3(a,c,d)C** was >95% and above 80% for (±)-**1b**.

Kinetic resolutions of these racemic amines (±)-**1a-d** with the investigated activated acylating agents **2A**-**C** have already been performed with various immobilized forms of CaLB, either via the covalent attachment or adsorption in various matrices such as mesoporous beads (CaLB T2-150), phenyl functionalized silica gel (CaLB G250P), and microporous acrylic resin (CaLB N435) [14,15,16,17]. In Table 1, the results of CaLB-MNPs and N435 in this work are compared with the relevant results of preceding studies.

The parameters analyzed were the conversion, enantiomeric excess (*ee*) of the formed amide (*R*)-**3(a**–**d)(A**–**C)** and residual amine (*S*)-**1a**–**d** fractions, and specific biocatalytic activity (*U*_B_) in the CaLB-MNP- and CaLB N435-catalyzed KRs. Although the enantiomeric excess of the product (*ee_p_*) is an important parameter for irreversible kinetic resolution, this alone cannot represent the degree of enantiomer selectivity (*E*) [40,41]. Since *E* is a constant independent of time and substrate concentrations controlled only with the ratio of the specific rate constants for the two enantiomers, *k*_R_/*k*_S_, this is an intrinsic measure of selectivity. In KRs with lower selectivity, the 50% limit of conversion could be exceeded; therefore, *ee_p_* should undergo a decrease. Thus, in a KR catalyzed using a biocatalyst of higher activity but lower selectivity, the product’s ee (*ee_p_*) should decrease above 50% conversion.

Data of Table 1 indicate that with acylating agents **2A** and **2B**, the sixth hour of the MNP reactions was comparable to the first hour of the reactions utilizing N435 in terms of conversion, while for KRs with **2C**, the first hour of both reactions was used in the comparison. In all cases, the *ee* of the product and the *E* values were equal or higher when utilizing the MNP-bound form of CaLB as compared to the forms immobilized onto porous carriers (N453 [15], and this work, Immozyme T2-150 [16], G250P [15,16]), indicating an enhancement in the selectivity of CaLB on MNPs compared to the N435 or other forms. The specific activity *U*_B_ values with **2C** were also comparable for CaLB-MNPs and N435 except in the KR of 1-phenylethan-1-amine (±)-**1c** where *U*_B_ of CaLB-MNPs remained significantly lower than that of N435.

An important note is that since the mass-to-mass ratio of CaLB and the MNP carrier was 1:15 in the immobilization with high immobilization yield, it could be calculated that approximately 0.8 mg of the CaLB/test was applied in this study. Although the quantity of CaLB in the N435 is not disclosed by the manufacturer, it could be estimated to be more than 10%, up to 2–3 mg [44]. This can explain why one mass unit of the CaLB-MNP was somewhat less active than the commercial N435, which is attributed to the approximate proportionality of the real CaLB content.

The diisopropyl malonate **2A** as an acylating agent was previously used in the kinetic resolution of racemic amines (±)-**1a**–**d** with the reference biocatalyst Novozyme 435 (N435; from Novozym: CaLB immobilized via adsorption within macroporous acrylic resin) [14]. The results of the reference literature listed as entries 3, 10, 17, and 29 in Table 1 show good conversion values (45–52.1%) with a high *ee* of the products (*R*)-**3(a**–**d)A** (92–99.9%) and good-to-excellent enantiomeric ratio/selectivity (>100–≫200), which are in good agreement with our reference data obtained with N435 using a lower concentration of the substrate (45 mM) and lesser quantities of biocatalysts (50 mg/mL) and acylating agent (1 equiv.), as shown in entries 2, 9, 16, and 28. The reactions carried out with the aid of CaLB-MNPs using diisopropyl malonate **2A** as an acylating agent under similar conditions are reported in entries 1, 8, 15, and 27.

Moderate conversion rates were obtained for (±)-**1a,c,d** (21.6–33.8%), which were comparable since half of the biocatalyst amount was utilized. For (±)-**1b**, a quite reasonable conversion (45.9%) was attained. The *ee* of the product was also excellent (99.0–99.6%), and the *E* value was on par with or superior to that of CaLB on macroporous supports with most of the investigated amines. Another advantage of the CaLB-MNP is that the filtration process to separate the biocatalyst can be eliminated due to the magnetic properties of the MNP-supported biocatalyst, enabling easy separation with magnetic decantation.

Isopropyl 2-cyanoacetate **2B** (and ethyl 2-cyanoacetate **2B***) were applied as acylating agents in another study on the kinetic resolution of (±)-**1c** with various immobilized forms of CaLB [15]. It was observed that the conversion values employing the two acylating agents with different natures of leaving alcohol did not differ significantly. The comparison of the three different immobilized forms of CaLB (CaLB N435, CaLB Immozyme T2-150, and CaLB G250P) with ethyl 2-cyanoacetate **2B*** (RT, in toluene-THF 1:1, for 24 h) revealed the best conversion with CaLB N435, and thus this form was chosen as the biocatalyst for subsequent studies. When using only 0.5 equivalent of the acylating agent, the conversions were only moderate after 24 h (31.7%, 20.0%, and 17.1%, respectively), as shown in entries 20, 22, and 23. The theoretical 50% conversion of a highly selective KR could only be achieved (50.1%), by using one equivalent of **2B*** (entry 21). In contrast to other results of this work indicating significantly higher enantiomer selectivity of CaLB-MNPs when using the reference form N435 in KRs of the other three amines (±)-**1a,c,d,** in KRs with 1 equiv. of **2B**, the selectivity in KR of 1-phenylethan-1-amine (±)-**1b** (entry 18) was similar to the one with N435 (entry 19), providing even a higher conversion rate (43.9%) and excellent *ee* of the product (*R*)-**3cB** (98.9%), which is comparable to the ones published (98.2–99.9%) [15].

Various 2-alkoxyacetates, such as isopropyl 2-ethoxyacetate **2C** for (±)-**1c** [16] and isopropyl 2-propoxyacetate for (±)-**1a**–**d** with [17], were applied as acylating agents in KRs of various amines in batch and continuous-flow (in a packed-bed reactor) modes. Isopropyl 2-ethoxyacetate **2C** (0.6 equiv.) was applied in the batch-mode KR of (±)-**1c** (385 mM) utilizing G250P (25 mg mL**^−1^**, CaLB immobilized into porous phenyl functionalized silica gel) to achieve a moderate conversion rate (33.8%) in 1 h at 30 °C, with excellent *ee*_(*R*)-**3cC**_ (>99.9%) (entry 26) [16]. With CaLB-CV-T2-150 (15 mg mL^−1^) and **2C** (1 equiv.) in KR of (±)-**1c** (778 mM), moderate conversion (26.1%) and excellent *ee*_(*R*)-**3cC**_ (99.6%) were reached in 1 h [17]. In the current study with CaLB-MNPs (50 mg mL^−1^) and **2C** (1 equiv.) in the KR of (±)-**1c** (45 mM) (entry 24), good conversion (40.2%) with comparable *ee*_(*R*)-**3cC**_ (99.4%) and *E* value (≫200) could be reached in 1 h.

### 3.3. CaLB-MNP-Catalyzed Kinetic Resolution of Chiral Amines (±)-**1b** and (±)-**1c** with Isopropyl 2-Ethoxyacetate **2C** in Thermostatted Continuous-Flow U-Shape Reactor

The KRs of drug-like racemic alcohols containing a nonplanar heterocyclic part such as 4-(morpholyn-4-yl)butan-2-ol [8], 4-(3,4-dihydroisoquinoline-2(1*H*)-yl)butan-2-ol, and 4-(3,4-dihydroquinoline-1(2*H*)-yl)butan-2-ol [9] utilizing CaLB-MNP catalysis in a simple and convenient continuous-flow U-shape reactor inspired us to implement this methodology for several racemic amines to expand the scope of substrates analyzed in this reactor. For this investigation, the KRs of the industrially important chiral amines, (*±*)-1-methoxy-2-propylamine (*±*)-**1b** and (*±*)-1-phenylethan-1-amine (*±*)-**1c**, were selected with the most efficient isopropyl 2-ethoxyacetate **2C** acylating agent in the *N*-acylation of racemic amines.

The continuous-flow reactions in a thermostatted continuous-flow U-shape reactor were carried out as detailed in Section 2.2.7. To conduct the continuous-flow experiments under industrially more relevant conditions, they were performed at an elevated concentration (100 mM) of the substrates (*±*)-**1b**,**c** at a 1 µL min^−1^ flow rate at 40 °C for over two days (Figure 3).

The continuous-flow KR of 1-methoxy-2-propylamine (±)-**1b** reached a stationary state after 4 h, which could be maintained until 22 h. In the continuous-flow KR of 1-phenylethan-1-amine (±)-**1c**, the stabilization took a bit longer than 4 h, but the conversion also started to decrease after 22 h. Due to activity requirements of the enzyme, the solvent could not be completely dried; therefore, the inhibition by 2-acetoxyacetic acid formed in parallel hydrolysis is the most plausible explanation for this phenomenon. Nevertheless, after 24 h, washing with a substrate-free solvent with a flow rate of 5 μL min^−1^ for 1 h could regain the original activity of CaLB-MNP, proving the reusability of CaLB-MNPs. After the 48th hour, the same behavior was observed, indicating that the washing step with a substrate-free solvent is necessary after each 24-h cycle. These results improve our understanding of this potent biocatalyst and corroborate previous findings about the reusability and recyclability of CaLB-MNPs [8,9].

In a previous report utilizing isopropyl 2-ethoxyacetate **2C** (0.6 equiv.) for the KR of (±)-**1c** (385 mM) in a packed-bed reactor (PBR) filled with CaLB-G250P under continuous-flow conditions (30 °C, 0.4 mL min^−1^ in toluene), good results were attained at (*c =* 31.9% with *ee*_(*R*)-**3cC**_ (>99.9%) [16]. Later, a PBR filled with CaLB-CV-T2-150 and isopropyl 2-propoxyacetate (0.6 equiv.) was used for continuous-flow KRs of (±)-**1a**–**d** (630 mM, in toluene), resulting in conversions ranging from 46% to 49% with high selectivity (*ee*’s of products 99.1–99.9% and *E* values ≫200) [17].

The results of continuous-flow reactions with other CaLB forms in PBRs cannot be directly compared with the results of the present study’s U-shape reactor as a comparison of the specific reaction rate (*r*) for a batch process (*r*_batch_; essentially identical to *U*_B_) with a flow process (*r*_flow_) is only possible under similar conditions at nearly identical conversions [45]). Therefore, the ratio of the specific reaction rate for a batch and the corresponding flow process (*r*_flow_*/r*_batch_) should be considered. The degree of enhancement was up to *r*_flow_*/r*_batch_ ~2 when a PBR was used in continuous-flow mode compared with the reaction under a batch mode [45]. Regarding the specific reaction rate values (*r* = *U*_b_) in KR with **2C** catalyzed using CaLB-MNPs for compound (±)-**1b**, an enhancement of *r*_flow_ = 9.6 U g^−1^ over *r*_batch_ = 7.3 U g^−1^ was noted, while for compound (±)-**1c**, a slightly decreased *r*_flow_ = 4.6 U g^−1^ from *r*_batch_ = 6.0 U g^−1^ was observed. These results indicate that except for the benefit of the continuous retention of the MNP-based catalyst, the U-shape reactor does not offer further advantages over batch-mode reactors. However, other types of MNP-based flow reactors offer the possibility of larger enhancement, which could be the subject of further studies. In a recent study on agitated MNP-based flow reactors, where different types of continuous-flow reactors were compared, including a cell corresponding to a cell of the U-shape reactor used in this work, the *U*_b_ value in a magnetically fixed cell could be enhanced further by three to four times in an agitated MNP flow reactor cell with rotational movement of two magnets (AFR_RM_) [46]. This enhancement of up to fourfold, achievable only with MNP-bound forms, significantly exceeds the maximum twofold enhancement found thus far in PBRs as compared to batch reactions.

The results show that using CaLB-MNPs in batch mode offers identical or superior selectivity with somewhat lower productivity than CaLB on porous supports. On the other hand, with the most efficient isopropyl 2-ethoxyacetate **2C**, CaLB-MNPs in the U-shape reactor proved to be efficient acylating agents, resulting in good conversion rates, and high-to-excellent enantiomer selectivity. The benefits of CaLB-MNP biocatalysts can be assessed in the future with other compounds of this class, making them a viable alternative for facilitating the kinetic resolution of chiral amines with the acylating agents studied. Future research can also be carried out with other types of acylating agents and flow reactors.

## 4. Conclusions

Generally, the kinetic resolution reactions of four racemic amines (±)-**1a-d** with three different acylating agents, namely diisopropyl malonate **2A**, isopropyl 2-cyanoacetate **2B**, and isopropyl 2-ethoxyacetate **2C**, were successfully performed with CaLB-MNPs as biocatalysts, with moderate-to-excellent conversion rates, providing high enantiomer selectivity in most cases. Although the processes catalyzed using the reference form of CaLB (N435) were usually more productive, in several instances, their degree of enantiomer selectivity remained lower than the CaLB-MNP form. Overall, isopropyl 2-ethoxyacetate **2C** performed the best for all amines (±)-**1a**–**d**, considering the balance between conversion values and enantiomer excess in KRs with the CaLB-MNP as well as with the reference N435 form. Besides the higher selectivity in several KRs, another benefit of using MNPs as CaLB supports is the facilitated purification and reuse processes. The results of continuous-flow experiments corroborate previous findings about the reusability and recyclability of CaLB-MNPs and improve our understanding of this potent biocatalyst in U-shape reactors or other MNP-based flow reactors.

## Data Availability

Not applicable.

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
