# Peer review of "Immobilization of Lipase B from Candida antarctica on Magnetic Nanoparticles Enhances Its Selectivity in Kinetic Resolutions of Chiral Amines with Several Acylating Agents"

_life, 2023, doi:10.3390/life13071560_

Round 1

Reviewer 1 Report

1.     Write a clear problem statement, research question, and research rationale to demonstrate the originality of this paper relative to the current state of the art.

2.     Several paragraphs have differing font sizes/styles (letters), and spaces are missing in various locations throughout the manuscript, so I would like to encourage the authors to proofread the entire document

3.     Authors should emphasize the originality of this study's concept by citing recent research. Otherwise, it is neither innovative nor sensible in the perspective of Journal readers

4.     It is suggested to include a comparison of the similarities and distinctions between this research and previous findings.

5.     The entire manuscript must be reviewed to prevent the repetition of the same information

6.     All abbreviations should be spelled out the first time they are used, followed by the abbreviation, and then only the abbreviation

7.     In addition, the authors should designate where future research efforts should be concentrated.

8.     The entire manuscript must be thoroughly edited for typographical and grammatical errors.

9.     Figures are of very poor quality. The legends are difficult to understand. Pls provide high quality images.

10.  Statistic part is completely missing in methodology section.

minor english required. 

Author Response

Reviewer 1: Comments and Suggestions for Authors

  1. Write a clear problem statement, research question, and research rationale to demonstrate the originality of this paper relative to the current state of the art.

Response: Additions were made in the abstract and in the introduction section to improve the clarity of the research goal and emphasize the originality of the paper.

  1. Several paragraphs have differing font sizes/styles (letters), and spaces are missing in various locations throughout the manuscript, so I would like to encourage the authors to proofread the entire document

Response: This problem may have appeared when using the Word version of the manuscript instead of the PDF, especially when handling the Word file under different operational systems (the authors have experienced similar problems previously between Mac and Windows, for example). Nevertheless, the revised manuscript was thoroughly proofread.

  1. Authors should emphasize the originality of this study's concept by citing recent research. Otherwise, it is neither innovative nor sensible in the perspective of Journal readers

Response: Although the original manuscript had more than 60% of its references from the last five years including pertinent research in the area of kinetic resolution, magnetic nanoparticles and continuous-flow reactors, a new paragraph was added to the Introduction to emphasize recent researches in the area: “Apart from the utilization in KR or DKR of enantiomer compounds, CaLB-MNPs are also used in biodiesel synthesis [34,35], transesterification reactions [36], or even utilized in a magnetic crosslinked inclusion body form [37].” To highlight the originality of this study concept proving its innovation, an additional sentence was added to the last paragraph of the Introduction, as it follows: “Although CaLB has been recently studied immobilized onto non-porous MNPs—having advantages over porous supports—there is a lack of studies on their behavior in KR of racemic amines.”

  1. It is suggested to include a comparison of the similarities and distinctions between this research and previous findings.

Response: In the sub-section 3.2 of Results and Discussion, the Table 1 comprises a vast comparison of CaLB-MNPs and CaLB N435-catalyzed batch-mode KRs of the racemic amines studied with the same acylating agents found in the literature, followed by an extensive discussion with similarities and distinctions. In sub-section 3.3, the comparisons are made focusing on continuous-flow reactors, and this work is contrasted with other works that utilized packed bed reactors (PBR) and agitated flow reactors with rotational movement (AFRRM). Considering the extensions for answering points 1, 3, and 7 we believe that novelties of the study are clearly defined.

  1. The entire manuscript must be reviewed to prevent the repetition of the same information

Response: The revised manuscript was extensively reviewed to prevent the repetition of the same information as suggested.

  1. All abbreviations should be spelled out the first time they are used, followed by the abbreviation, and then only the abbreviation

Response: In the proofreading process of the revised manuscript, all the errors concerning this issue were corrected.

  1. In addition, the authors should designate where future research efforts should be concentrated.

Response: The last paragraph of Results and Discussion was rewritten to make it clear the possibility of future research as follows: “The benefits of the CaLB-MNP biocatalyst can be assessed in the future with other compounds of this class, making it a viable alternative for assisting the kinetic resolution of chiral amines with the acylating agents studied. Future research can also be done with further types of acylating agents and flow reactors.”

  1. The entire manuscript must be thoroughly edited for typographical and grammatical errors.

Response: The revised manuscript was rigorously edited for typographical and grammar mistakes, as Reviewers 3 and 4 suggested.

  1. Figures are of very poor quality. The legends are difficult to understand. Pls provide high quality images.

Response: We reedited Figure 1 for better visibility. In addition, we converted Figure 2 from the normal widths to full widths, representing an almost 30% enlargement of each panel, which made the graphs and their legends more visible and readable. Also, all the images were uploaded with at least 300 dpi in the editorial system.

  1. Statistic part is completely missing in methodology section.

Response: We added a paragraph sub-section 2.2.4 of Materials and Methods explaining the situation with statistics: “A variance of less than 2% was found in the preliminary activity test in batch mode, performed in triplicates. Therefore, the optimization and time course experiments were made as single series.”

Reviewer 2 Report

The limitation of the work must be mentioned in introduction section

Abstract: It is necessary to include a succinct and factual abstract that briefly outlines the study's objectives, key findings, and important recommendations.

Author Response

Reviewer 2: Comments and Suggestions for Authors

  1. The limitation of the work must be mentioned in introduction section

Response: The introduction part was slightly modified taking into consideration the similar notes of other reviewers, and the limitation of the work was mentioned. We believe that his extension of the manuscript improved the quality and clarity of the work.

  1. Abstract: It is necessary to include a succinct and factual abstract that briefly outlines the study's objectives, key findings, and important recommendations.

Response: The abstract was changed including the objectives, key findings, and recommendations of this work. We believe that the modifications made in the Abstract and in the Introduction section could clarify the originality and innovation of this work, as Reviewer 1 also suggested.

Reviewer 3 Report

The study reports the kinetic resolution of several chiral amines was catalyzed by a biocatalyst obtained by covalent immobilization of lipase B from Candida antarctica on magnetic nanoparticles (CaLB-MNPs). Additionaly, utilizing different isopropyl esters as acylating agents, a simple, but effective and convenient MNP-based PTFE tube microreactor with six adjusting permanent magnets underneath the tube was used for the enantiomer selective acylation of drug-like racemic alcohols. However, there are several issues that need to be revised before you can reconsider accepting this article for publication.

1. Is there a limit to the number of keywords in the journal? In my opinion, I think the number of keywords chosen by the author is too large and should be appropriately reduced to highlight the main points.

2. The authors should add relevant characterization images of CaLB-MNPS, such as TEM and SEM images, to prove that CaLB is successfully immobilized on MNPs.

3. There are many errors in the article that do not match the language and the figures. For example, in the line 328 and 329 of the article, the corresponding data of the article should be Figure 2 instead of Figure 1.

4. In the supporting documents, Tables S1, S2 and S3 only use different acylation agents. The data of the three tables can be organized and put in one table to make the comparison of the results more clearly visible.

5. The Resources section also has a lot of formatting problems. For example, in reference 4, the Volume code format should be consistent with other references, and the "volume" should be deleted. The format of reference 25 does not correspond to other references at all, please check again carefully.

There is no logic problem in the English expression as a whole, but there are many small grammatical mistakes, please check and correct them carefully.

Author Response

Reviewer 3: Comments and Suggestions for Authors

The study reports the kinetic resolution of several chiral amines was catalyzed by a biocatalyst obtained by covalent immobilization of lipase B from Candida antarctica on magnetic nanoparticles (CaLB-MNPs). Additionaly, utilizing different isopropyl esters as acylating agents, a simple, but effective and convenient MNP-based PTFE tube microreactor with six adjusting permanent magnets underneath the tube was used for the enantiomer selective acylation of drug-like racemic alcohols. However, there are several issues that need to be revised before you can reconsider accepting this article for publication.

  1. Is there a limit to the number of keywords in the journal? In my opinion, I think the number of keywords chosen by the author is too large and should be appropriately reduced to highlight the main points.

Response: The template suggests using “three to ten pertinent keywords specific to the article yet reasonably common within the subject discipline.” To appropriately reduce the number of keywords and highlight the main points as suggested, the names of the three acylating agents, actual compounds, were removed, decreasing from ten to seven keywords.

  1. The authors should add relevant characterization images of CaLB-MNPS, such as TEM and SEM images, to prove that CaLB is successfully immobilized on MNPs.

Response: According to the note of Reviewer Scanning Electron Microscope (SEM) images were made from the empty MNPs as well as from the CaLB-MNPs.        
We have added the following paragraphs for the revised manuscript: subsection 2.2.3. Preparation of CaLB-MNPs biocatalysts section: “The surface morphology of the samples was investigated with a JEOL JSM-5500LV scanning electron microscope (SEM). An electron beam energy of 15 kV was used for the investigations. The SEM images are given in Fig. S1 in the Supplementary Material.”      
Analysis of the experimental results explaining the successful immobilization was also made in Results and discussion as a full new sub-section 3.1. Biocatalyst characterization.

TEM images, IR and Raman investigations of the CaLB-MNPs are expected to be made as part of our forthcoming study.

  1. There are many errors in the article that do not match the language and the figures. For example, in the line 328 and 329 of the article, the corresponding data of the article should be Figure 2 instead of Figure 1.

Response: The revised manuscript was thoroughly proofread and edited for typographical and grammar mistakes, as also suggested by Reviewers 1 and 4.

  1. In the supporting documents, Tables S1, S2 and S3 only use different acylation agents. The data of the three tables can be organized and put in one table to make the comparison of the results more clearly visible.

Response: The data of the three tables were organized and comprised as one for more clear and visible results. Thank you for your input.

  1. The Resources section also has a lot of formatting problems. For example, in reference 4, the Volume code format should be consistent with other references, and the "volume" should be deleted. The format of reference 25 does not correspond to other references at all, please check again carefully.

Response: Reference 4 is a book chapter, and it was formatted according to the template provided by Life. Reference 25 is a patent, and since there was no indication of how to format it, we tried to approximate the reference to the others used. Nevertheless, all references were checked again carefully to comply with the corresponding formats.

Reviewer 4 Report

This is a good study that reports on the feasibility of utilizing nanoparticle-immobilized lipase for the purpose of kinetic resolution of chiral amines.  The results have implications at the industrial level to achieve selectivity and high levels of resolution with very high enantiomeric excess values.   Among the main advantages are the ease of separation, the potential for flow reactor implementation and the higher efficiency.  I do have a few requests that the authors need to answer:

1. Figure 1 shows a depiction of the CaLB-MNPs reactor.  It would be useful if the authors added a picture of an actual unit with the syringe for reagent injection on it.

2. The experimental section is appropriate, but the mass spectra results are required to report the structure of the products.

3. Line 328 refers to Fig.1 but it should refer to Fig. 2

4. The plots comparing the commercial product N435 with the CaLB-MNPs are very interesting in that the latter performs far better over time.  For example figure 2e shows a slight decay in the ee whereas figure 2f shows almost complete decay with the N435 product. 

5. Can the authors explain the process by which the ee decays over time and how it relates to the efficiency of the nanoparticles?

6. Is the enantiomeric selectivity / ratio needed here?  I am not even sure what that parameter really mean and it is probably quite novel.  Maybe they should add an explanation in line 335.

7. Can the authors explain why the nanoparticles react somewhat slower than N435 but yet in a kinetically simpler manner?  This is also more convenient for process development.

8. Please explain the sudden decay in the conversion percentage observed at the end of Figure 3.  Also what was the ee for this case and was the ee also decreasing with time?  Would that preclude using the CaLB-MNPs as flow reactors?

Author Response

Reviewer 4: Comments and Suggestions for Authors

This is a good study that reports on the feasibility of utilizing nanoparticle-immobilized lipase for the purpose of kinetic resolution of chiral amines.  The results have implications at the industrial level to achieve selectivity and high levels of resolution with very high enantiomeric excess values.   Among the main advantages are the ease of separation, the potential for flow reactor implementation and the higher efficiency.  I do have a few requests that the authors need to answer:

  1. Figure 1 shows a depiction of the CaLB-MNPs reactor.  It would be useful if the authors added a picture of an actual unit with the syringe for reagent injection on it.

Response: A picture of an actual unit of the reactor with the syringe for reagent injection and the thermostat used was added to the Supplementary Materials. A statement is also added to sub-section 2.2.6 of Materials and methods: “Pictures showing the entire reactor setup are presented in Fig. S2 in Supplementary Material.”

  1. The experimental section is appropriate, but the mass spectra results are required to report the structure of the products.

Response: The three novel compounds were characterized by GC, NMR, and IR. The other compounds were previously described in the literature and characterized by the same way.

  1. Line 328 refers to Fig.1 but it should refer to Fig. 2

Response: The revised manuscript was thoroughly proofread and edited for typographical and grammar mistakes, as also suggested by Reviewer 1 and 3.

  1. The plots comparing the commercial product N435 with the CaLB-MNPs are very interesting in that the latter performs far better over time.  For example figure 2e shows a slight decay in the ee whereas figure 2f shows almost complete decay with the N435 product.

Response: This is exactly the conclusion of these experiments, where utilizing CaLB-MNPs and isopropyl 2-ethoxyacetate 2C as acylating agent (Fig. 2e) in comparison with utilizing the commercial product N435 (Fig. 2f) a better balance between conversion and enantiomer selectivity is achieved. This and the extensive discussion with comparisons with applications reported in the literature can prove that CaLB-MNPs are potent biocatalysts.

  1. Can the authors explain the process by which the ee decays over time and how it relates to the efficiency of the nanoparticles?

Response: An explanation for this question together with question 6 was added in the main article. “Although the enantiomeric excess of the product (eep) is an important parameter for irreversible kinetic resolution, this alone cannot represent the degree of enantiomer selectivity (E) [40,41]. Since E is a constant independent of time and substrate concentrations controlled only by the ratio of the specific rate constants for the two enantiomers, kR/kS, this is an intrinsic measure of the selectivity. In KRs with lower selectivity, the 50% limit of conversion could be exceeded, therefore eep should have a decay. Thus, in a KR catalyzed by a biocatalyst of higher activity but lower selectivity, the product’s ee (eep) should decrease above 50% conversion.”

  1. Is the enantiomeric selectivity / ratio needed here?  I am not even sure what that parameter really mean and it is probably quite novel.  Maybe they should add an explanation in line 335.

Response: Enantiomeric ratio (E) was defined by Chen et. al. in 1982 as an important biochemical parameter [40]. It is a constant independent of time and substrate concentrations controlled by the ratio of the specificity constants, V/K, where V corresponds to maximal velocity and K to Michaelis constant, which are intrinsic properties of that biochemical system. Considering this may not be clear to all the readers, a definition (E= kR/kS) and an interpretation of the values were added.

  1. Can the authors explain why the nanoparticles react somewhat slower than N435 but yet in a kinetically simpler manner?  This is also more convenient for process development.

Response: To explain this phenomenon, the discussion on the content of Table 1 was complemented with the following paragraph: “An important note is that since the mass-to-mass ratio of CaLB and the MNP carrier was 1:15 in the immobilization with high immobilization yield, it could be calculated that approximately 0.8 mg of the CaLB/test was applied in this study. Although the quantity of CaLB in the N435 is not disclosed by the manufacturer, it could be estimated to be more than 10%, up to 2-3 mg [44]. This can explain why one mass unit of the CaLB-MNP is somewhat less active than the commercial N435 and can be seen in the approximate proportionality of the real CaLB content.”

  1. Please explain the sudden decay in the conversion percentage observed at the end of Figure 3.  Also what was the ee for this case and was the ee also decreasing with time?  Would that preclude using the CaLB-MNPs as flow reactors?

Response: The decay was better explained with the addition the following parts highlighted by yellow to the discussion: “Due to activity reasons, the solvent cannot be completely dry; therefore, the inhibition by 2-acetoxyacetic acid forming in a parallel hydrolysis is the most plausible explanation for this phenomenon. Nevertheless, after the 24th hour, washing with a substrate-free solvent with a flow rate of 5 μL min−1 for 1 h could regain the original activity of CaLB-MNP, proving the reusability of the CaLB-MNPs. After the 48th hour, the same behavior is observed, indicating that the washing step with substrate-free solvent is necessary after each 24-hour cycle.”

The eeP for the acylated fraction of racemic 1-methoxypropan-2-amine (±)-1b [(R)-3bC] varied according to the answers for points 5 and 6, while in the KR of racemic 1-phenylethan-1-amine (±)-1c with significantly higher selectivity the eeP of the acylated fraction of [(R)-3cC] remained >98% constantly.

Round 2

Reviewer 1 Report

Authors has revised manuscript satisfactory. Hence suitable for publication. 

Authors has revised manuscript satisfactory. Hence suitable for publication.